# Time to Move to the Single-Cell Level: Applications of Single-Cell Multi-Omics to Hematological Malignancies and Waldenström’s Macroglobulinemia—A Particularly Heterogeneous Lymphoma

**DOI:** 10.3390/cancers13071541

**Published:** 2021-03-26

**Authors:** Ramón García-Sanz, Cristina Jiménez

**Affiliations:** Hematology Department, University Hospital of Salamanca (HUS/IBSAL), CIBERONC and Cancer Research Institute of Salamanca-IBMCC (USAL-CSIC), 37007 Salamanca, Spain; jscris@usal.es

**Keywords:** single-cell sequencing, applications, hematology, Waldenström’s macroglobulinemia

## Abstract

**Simple Summary:**

Intra-tumor heterogeneity is inherent to all cancers and makes direct mapping of genotype–phenotype relationships challenging. The advent of single-cell multi-omics techniques has allowed us to begin to comprehensively dissect cellular heterogeneity and access biological information unobtainable from bulk analysis. Applications cover many fields and are increasingly numerous. This review aims to summarize the most important applications of single-cell technologies in hematological tumors, providing a translational view. Data show the power of single-cell multi-omics to resolve the complex biology of heterogeneous populations and to derive information that can be used to improve treatment strategies. We discuss, with a practical example, how to make use of these techniques to study the heterogeneity of a specific type of monoclonal gammopathy called Waldenström’s macroglobulinemia.

**Abstract:**

Single-cell sequencing techniques have become a powerful tool for characterizing intra-tumor heterogeneity, which has been reflected in the increasing number of studies carried out and reported. We have rigorously reviewed and compiled the information about these techniques inasmuch as they are relative to the area of hematology to provide a practical view of their potential applications. Studies show how single-cell multi-omics can overcome the limitations of bulk sequencing and be applied at all stages of tumor development, giving insights into the origin and pathogenesis of the tumors, the clonal architecture and evolution, or the mechanisms of therapy resistance. Information at the single-cell level may help resolve questions related to intra-tumor heterogeneity that have not been previously explained by other techniques. With that in mind, we review the existing knowledge about a heterogeneous lymphoma called Waldenström’s macroglobulinemia and discuss how single-cell studies may help elucidate the underlying causes of this heterogeneity.

## 1. Introduction

Despite its monoclonal origin, intra-tumor heterogeneity (ITH) is inherent to the majority of cancers and can be associated with genetic variability, epigenetic modifications, gene and protein expression, metabolism, morphology, and other features of tumors [1]. Moreover, ITH is known to play a significant role in tumor progression, preservation of oncogenic potential, cell survival under changing microenvironmental conditions, and resistance to drug therapy [1,2,3], the latter being even more relevant nowadays due to the use of targeted therapies that may favor the selection and expansion of the clones [4,5]. Accurate evaluation of ITH could be essential for efficient diagnosis and for successful treatment and outcome [6]. Determining the pattern of somatic mutations is no longer sufficient, but it is essential to know their distribution in individual tumor cells and clones, and how they interact to influence tumor fitness [7]. Approaches based on molecular genetics have been developed to study tumor behavior and predict treatment efficiency by considering factors and mechanisms for generating ITH. Despite its widespread use and recognized diagnostic capacity, next-generation sequencing of the bulk tumor reveals only a minority of the genetic aberrations present in an entire tumor and can lead to the mutational burden being underestimated. In addition, it is not possible to distinguish which alterations occur in the same clone(s) or to elucidate the order of appearance and co-dependency of mutations. Single biopsies from geographically localized tumor areas cannot recapitulate the complexity of spatial heterogeneity. Such problems inherent to tumor sampling bias due to ITH may be mitigated by circulating tumor DNA (ctDNA) sampling [8,9]. Sequencing ctDNA can provide a wider vision of the tumor genomic landscape but can never reveal how certain genetic abnormalities change the behavior of the cell subset involved.

Single-cell sequencing techniques represent a step forward in the characterization of the complete tumor architecture, revolutionizing the landscape with respect to ITH. Not only that, single-cell genomics allows us to determine the timing of the alterations (early vs. late mutations) and provide information about mutations that cooperate in the development of the disease. Genomic, transcriptomic, and epigenomic assays at the single-cell level are transforming our understanding of cellular complexity, thereby enabling us to access biological information that cannot be obtained from bulk analysis [10].

Conventional single-cell methods are based on manual-picking micropipettes [11] or cell fluorescence-activated cell sorting (FACS) [12]. By contrast, the new methodologies include microfluidic technologies, such as the C1™ system from Fluidigm^®^ (South San Francisco, CA, USA), which isolates single-cells into individual reaction chambers in an integrated fluidic circuit [13], droplet-based microfluidic technologies, such as the Tapestri System (MissionBio, South San Francisco, CA, USA) [14], the Chromium System (10× Genomics, Pleasanton, CA, USA) [15], and the Cellular Indexing of Transcriptomes and Epitopes by Sequencing (CITE-seq) (Technology Innovation Lab, New York Genome Center, New York, NY, USA) [16], and nanowell-based technologies, such as the Rhapsody™ Single-Cell Analysis System (BD Biosciences, San Jose, CA, USA), which uses planar arrays of microwells for cell capture [17], and the Seq-Well platform (Shalek Lab, Massachusetts Institute of Technology and the Broad Institute, Cambridge, MA, USA) [18], to mention the most important ones.

Single-cell methods involve single-cell isolation, barcoding, and sequencing to analyze multiple types of molecules from individual cells, as well as the integrative analysis of the data to characterize cell types and their functions based on molecular signatures. To analyze multiple types of molecules from the same cell, it is essential to isolate the single cells and then to barcode the molecules. The process will be explained in greater details for two of the most widely used single-cell systems. The Tapestri Platform (Mission Bio) allows DNA and protein analysis of each individual cell and is based on a two-step microfluidic workflow. The first step involves encapsulating the cells into sub-nanoliter droplets and isolating DNA and oligo-conjugate antibodies from each single cell. The second step involves cell barcoding and targeted amplification by polymerase chain reaction (PCR) within the droplets, which are then disrupted, before extracting barcoded DNA for library amplification. Final libraries are purified and sequenced. The Chromium Platform (10× Genomics) enables the analysis of gene expression, chromatin accessibility, cell-surface proteins, immune clonotype, antigen specificity, and clustered regularly interspaced short palindromic repeats (CRISPR) edits. It is based on the generation of thousands of single-cell partitions, each containing an identifying barcode for downstream analysis using advanced microfluidics. Within the instrument, barcoded gel beads, coated with a unique oligonucleotide barcode sequence and functionalized sequences to capture molecules of interest, are mixed with cells or nuclei, enzymes, and partitioning oil to form thousands of single-cell emulsion droplets. Each droplet is the location of an individual reaction in which the beads are dissolved and molecules of interest from each cell are captured and barcoded. After barcoding, all fragments from the same cell or nucleus share a common barcode. Barcoded fragments from thousands of cells are pooled for downstream reactions to create sequencing libraries. After sequencing, bioinformatics tools use the identifying barcodes to map sequencing reads back to their single cell or nucleus of origin.

Although there are already many reviews of this topic, most of them have focused on the technical aspects of the technologies. Here, instead, we have addressed their practical aspects, presenting a translational point of view. This is a powerful, emerging technique, as evinced by the increasing rate at which studies are being published that demonstrate its broad range of applications. This review confines itself to considering the applications in the field of hematology to provide a comprehensive view of the potential value of these technologies. It also discusses how these applications can be translated to the study of Waldenström’s macroglobulinemia, a particular type of lymphoma/monoclonal gammopathy that is a good example of ITH. Finally, the existing challenges and limitations are considered.

## 2. Applications of Single-Cell Multi-Omics in Hematological Malignancies

Single-cell analysis has the potential to contribute to our understanding of cell complexity, and, specifically, there are many areas of hematology where it could be of great interest (Table 1). We consider the most relevant of these below.

### 2.1. Characterizing Immune Cell Populations

Single-cell RNA-sequencing studies, with different objectives, have yielded descriptions of a wide range of immune cells. For example, the characterization of human hematopoietic populations at the earliest stage of hematopoiesis (i.e., CD34+ cells) could provide a basis for generating blood and immune cells for clinical purposes [19], the study of CD4+ T-cells allows the specific cell states that drive disease or treatment response to be identified [20], and the analysis of stem cell-like CD8+ memory T-cells may help in the development of immunotherapies and vaccines [21].

### 2.2. Defining the Transcriptomic, Proteomic, and Epigenomic Identity of Malignant Cells

Genotyping and single-cell RNA-sequencing can be integrated to determine how somatic mutations corrupt the hematopoietic process. Nam et al. showed that malignant hematopoietic progenitors (CD34+ cells) with mutated *CALR* have a fitness advantage with respect to myeloid differentiation and upregulation of the nuclear factor kappa B (NF-κB) pathway, highlighting the cell-identity dependency of somatic mutations in human hematopoiesis [22].

Single-Cell DNA and Antibody Sequencing (DAb-seq) is a method that integrates DNA profiling and surface proteins of single cells at high throughput, allowing the proteogenomic dynamics of multiple patients to be tracked over multiple treatments and recurrences. Analyzing leukemia patients using this approach identified extensive genotype–phenotype decoupling, with immunophenotypic heterogeneity among cells with the same pathogenic mutation, as well as genotypically diverse cells with a convergent malignant immunophenotype, suggesting that independent phenotype or genotype measurements do not adequately capture the full extent of proteogenomic heterogeneity [23].

A different single-cell approach, established by Granja et al., combines highly multiplexed protein quantification, transcriptome profiling, and analysis of chromatin accessibility to deconvolve aberrant molecular features within blood from patients with mixed-phenotype acute leukemia, with the aim of identifying the causes of the disease. This integrative analysis made it possible to infer that the transcription factor *RUNX1* acts as a potential oncogene, regulating malignant genes associated with poor survival [24].

### 2.3. Detecting Pre-Leukemic Clonal Hematopoiesis Mutations

Another interesting application is the discrimination of mutations associated with age-related clonal hematopoiesis vs. true leukemia, in this case, acute myeloid leukemia (AML) [25]. Mutations in epigenetic regulators *DNMT3A* and *TET2* are frequently found in older individuals and appear to be associated with an increased death risk [26,27]. These mutations are also common in patients with AML [28], making genomic measurement of residual disease very difficult, since detected variants cannot be confidently assigned to the malignancy (particularly in the post-treatment setting) [29,30]. While targeted DNA sequencing cannot determine whether mutations associated with clonal hematopoiesis are also represented within the malignant clone, detection of fusion genes by single-cell DNA and antibody-oligonucleotide sequencing can resolve the leukemic clonal architecture, thereby enabling the comprehensive assessment of AML.

The presence of clonal hematopoiesis has also been assessed by single-cell DNA sequencing in patients with myelodysplastic syndrome-associated phenotypic abnormalities, which are present in ∼10% of newly diagnosed multiple myeloma (MM) cases [31].

### 2.4. Establishing the Order of Events That Mediate Cancer Origin and Evolution

A comprehensive understanding of disease development may facilitate the rational design of antitumor drugs and prevention strategies. Single-cell sequencing has made it possible to measure the clonal structures of childhood acute lymphoblastic leukemia (ALL) samples at diagnosis, and to establish the sequence of genetic events that underlie the disease; first, most of the structural variants, followed by single-nucleotide variants (SNVs), with *KRAS* (Kirsten Rat Sarcoma Viral Oncogene Homolog) mutations occurring late in disease development [32]. The order of mutation acquisition and the progenitor cells in which this process is initiated have also been studied in T-cell ALL (T-ALL). Patients showed an early event in a known oncogene (*MED12*, *STAT5B*) that was already detectable in multipotent progenitor cells. Intermediate events included copy-number alterations (such as loss of 9p21) and acquisition of fusion genes, while *NOTCH1* mutations were typically late events [33], despite having a key role in the development of T-ALL [34].

Reconstructing the complete clonal architecture of the tumor may give insights into its evolution and inform decisions about treatment [35,36]. Myeloid malignancies, including AML, arise from the expansion of clonal hematopoietic stem and progenitor cells that acquire subsequent somatic mutations. Single-cell sequencing has allowed clonal trajectories to be mapped and synergistic combinations of mutations to be revealed that promote clonal expansion and dominance. The work of Miles et al. demonstrated that clonal complexity increased as the disease progressed (from clonal hematopoiesis to AML), and continued to evolve as AML clones acquired mutations in signaling effectors, which were often sub-clonal and not concurrent. In addition, mutational combinations contributed to clonal dominance in various ways: specific co-occurring disease alleles (e.g., *NPM1*–*FLT3*-*ITD* or *DNMT3A*–*IDH2*) were associated with clonal dominance, whereas other mutational combinations (e.g., *NPM1*–*RAS*) did not promote clonal expansion. Changes in clonal architecture were due to the expansion of pre-existing minor clones and could be detected using single-cell DNA sequencing, drawing attention to the value of searching for therapies targeting these clones before they achieve clonal dominance. Finally, by combining mutational analysis with protein expression, it is possible to identify significant genotype-driven changes in cell-surface protein expression (e.g., mutations in the mitogen-activated protein kinase/extracellular signal-regulated kinases, MAPK/ERK, pathway that led to increased CD11b expression) [37]. Evidence of molecular complexity including intra-lineage clonal evolution was found in another study of myeloproliferative neoplasms: two unusual patients, each harboring two driver mutations (*JAK2*/*CALR* and *JAK2*/*MPL*), which represented, in both cases, independent clones. These data highlight the importance of routine assessment of the three canonical driver mutations (i.e., *JAK2*/*CALR*/*MPL*) in order to characterize disease biology accurately [38].

### 2.5. Describing Mechanisms That Lead to Disease Progression and Resistance to Therapy

Much research has been done to understand the underlying etiology of clonal evolution. This helps to determine whether sequential molecular monitoring is of clinical value for the early detection of malignancies, and to assess the utility of preventive interventions. Single-cell techniques have been used to study clonal evolution of severe aplastic anemia to a secondary myeloid malignancy. DNA sequencing at the single-cell level showed that sequential mutation acquisition over a 7-year clonal period occurred within the same clone derived from an initial *ASXL1* mutated clonal population [39]. The same strategy was used to study the predisposition to develop leukemia in Shwachman-Diamond syndrome (SDS). The authors linked the development of leukemia to the acquisition of biallelic *TP53* alterations that drove progression of *TP53* mutated clones. As SDS patients can develop multiple independent *TP53* mutated clones, serial monitoring by bulk sequencing would fail to distinguish clinically significant sub-clonal changes in *TP53* allelic status. These findings imply the value of integrating of single-cell DNA sequencing into surveillance strategies in order to identify patients with high-risk clones [40]. In MM, single-cell transcriptome analysis revealed that extramedullary progression is associated with alterations in the transcriptional programs of plasma cells and the microenvironment affecting proliferation and immune evasion [41].

Regarding the development of drug resistance, tumors harbor a variety of cell types that are thought to play an important role in this process [42]. Thus, in patients with AML, a single-cell study observed that greater clonal complexity was associated with reduced elimination of all malignant clones with standard chemotherapy regimens, and therefore with a higher risk of resistant clones persisting and eventually causing clinical relapse [43]. Therefore, an important prognostic application would be one that established a diversity index for the tumor, or parts of it, that could predict whether a patient will respond poorly to therapy or have a strong tendency to relapse. Cell composition could also determine a worse clinical outcome. Single-cell RNA-sequencing identified the CD34^+^CD117^dim^ proportion as an independent factor of poor prognosis in patients with t(8;21) AML, and this leukemic cluster was able to expand at the post-relapse refractory stage after several cycles of chemotherapy [44]. In addition, it is also possible to identify the molecular determinants of clinically relevant outcomes. In patients with AML who are treated with combinations based on venetoclax (a BCL-2 inhibitor), *NPM1* and *IDH2* mutations are associated with high response rates and durable remissions, while the activation of the FLT3, RAS, and TP53 signaling pathways seems to be linked to resistance development [45]. Emergence of multiple clones, each with distinct mechanisms of resistance, is a common finding following the secondary failure of single-agent targeted therapies for leukemias. An integrated genomic analysis combining DNA sequencing, RNA-sequencing, and a methylation profiling microarray revealed that selection of co-occurring mutations in hematopoietic transcription factor genes (*RUNX1*/*CEBPA*) or RAS-RTK (receptor tyrosine kinase) pathway genes were the main drivers of acquired resistance to IDH inhibitors, suggesting that novel strategies targeting certain high-risk co-occurring mutations might improve the therapeutic efficacy of IDH inhibitors in AML [46]. The complex biology of resistance was also highlighted in AML treated with ivosidenib (an inhibitor of mutated IDH1), whereby the concurrence of different mechanisms, particularly receptor tyrosine kinase pathway mutations and *IDH*-related mutations, contributed to primary and secondary resistance [47]. Likewise, McMahon and colleagues demonstrated that secondary clinical resistance to the FLT3 inhibitor gilteritinib in relapsed AML was commonly mediated by heterogeneous mutations that activate downstream RAS-MAPK pathways [35]. RNA-sequencing identified a CD19 splice variant as a biomarker of the failure of blinatumomab (anti-CD19/CD3) therapy in B-cell progenitor ALL [48], and a *USP7* gene signature was associated with resistance to therapy in AML patients [49]. Lastly, a deeper analysis of the CLL response to ibrutinib (a BTK inhibitor) allowed the dissection of the exact cellular and molecular changes induced by this drug and identified candidate molecular markers of therapy response. Chromatin accessibility analysis with ATAC-Seq on FACS-purified immune cell populations was combined with single-cell RNA-sequencing for a subset of timepoints. The high level of detail and biological insight obtained make it particularly well-suited to applications in personalized medicine, whereby each patient may follow a different disease trajectory, and for early-stage clinical trials of new targeted therapies, in which it is critical to understand the molecular and cellular dynamics in order to establish the appropriate dose and identify response biomarkers [50].

### 2.6. Studying the Microenvironment

The bone marrow microenvironment may support tumor progression and treatment evasion. Witkowsky et al. used single-cell RNA-sequencing and cellular indexing of transcriptomes and epitopes by sequencing (CITE-seq) to show that B-cell ALL remodels the bone marrow immune microenvironment upon disease initiation, as well as the subsequent recovery during conventional chemotherapy [51]. In addition, immune microenvironment profiling identified extrinsic regulators of survival, which would underpin the new immune-based therapeutic approaches for high-risk B-ALL treatment [51].

### 2.7. Understanding Human B-Cell Biology and Lymphoma Pathology by Modeling the Germinal Center 

Integrative single-cell analysis of gene expression, surface phenotype, and the B-cell receptor sequence can be useful for studying the heterogeneous germinal center (GC) B-cell subset [52]. This approach was used to characterize B-cell transcriptional heterogeneity in follicular lymphoma (FL) and compare it with the various states observed in normal GC B-cells. Results showed a major desynchronization of GC-specific gene-expression programs in FL tumor cells, although with distinct FL-specific cell states coexisting within a single patient. This suggests that lymphoma B-cells are not blocked in a GC B-cell state but might adopt dynamic modes of functional diversity [53].

### 2.8. Cancer Therapy

Transferring T-cells that express tumor-reactive T-cell receptors (TCRs) can induce regression of tumors in patients with advanced cancer. However, isolation and expression of a tumor antigen-specific TCR is a highly complex process. Advances in single-cell sequencing have begun to streamline this process; in particular, single-cell variable (V), diversity (D) and joining (J) genes (VDJ) sequencing data are used to clone antigen-specific TCRs. The same strategy can be used to expand primary human antigen-specific T-cell clones [54].

A different application is the identification of transcriptomic features in anti-CD19 Chimeric Antigen Receptor T-Cell (CAR T)-infusion cell products. This has been associated with efficacy and toxicity in diffuse large B-cell lymphoma (DLBCL). Results have shown that heterogeneity of the cellular and molecular features contributed to the variation in efficacy and toxicity and that the 7-day molecular response might serve as an early predictor of CAR T-cell efficacy [55].

### 2.9. Other Applications

CRISPR-Cas9 (CRISPR-associated endonuclease Cas9) gene editing allows the rapid interrogation of the functional impact of somatic mutations in human cancers. Single-cell technologies enable the analysis of Cas9-introduced gene edits in thousands of cells, quantifying mutational co-occurrences and zygosity status. As most cancers arise and propagate due to the complex interaction of multiple drivers, these tools allow the contribution of individual genetic drivers to cellular fitness to be assessed and the mutational co-occurrences, interactions of multiple lesions, and clonal evolutionary mechanisms in cellular and mouse models to be studied [56].

Another interesting application is bone marrow engraftment monitoring by the assessment of donor/host chimerism using the individuals’ unique genotype signatures as genetic proxies. Using a droplet microfluidic approach in AML patients at different times (before/after bone marrow transplant and at AML relapse), it was possible to obtain the molecular profile of single nucleotide variants (SNVs) across thousands of cells. This revealed the existence of genetic chimerism after bone marrow transplantation. In addition, comparison of clone number and size across the three times suggested that AML relapse after transplant may result from the aggressive and exclusive expansion of the oncogenic cells carrying tumor-suppressor and/or oncogene mutation(s), and is associated with loss of donor chimerism [57].

### 2.10. Future Applications

Assays and methods to detect molecular abnormalities will soon be adapted for use at the single-cell level. Fold et al. demonstrated fusion event detection at single-cell resolution using barcoded single-cell RNA-sequencing data in MM, pointing the way forward for the development of fusion methods [58]. 

## 3. Single-Cell Techniques in Waldenström’s Macroglobulinemia: Utility and Applicability

As we have seen, single-cell technologies have proven useful in many fields. Now, we will focus on the specific applications that these techniques may have in a particularly heterogeneous hybrid disease. ITH may be the underlying cause of the wide multilevel heterogeneity present in Waldenström’s macroglobulinemia (WM). WM is a rare indolent B-cell lymphoproliferative disorder characterized by bone marrow infiltration by lymphoplasmacytic lymphoma and the presence of an immunoglobulin M (IgM) monoclonal component [59]. The cellular composition of this lymphoma includes malignant lymphocytes and plasma cells. At the clinical level, the disease is consistently heterogeneous, its variety of behaviors being manifested in entities ranging from indolent forms, such as IgM monoclonal gammopathy of undetermined significance (IgM-MGUS) and asymptomatic WM (AWM), to highly symptomatic disease (symptomatic WM, SWM) [59]. Its evolution is highly variable [60], and transformation into aggressive DLBCL as well as leukemia has also been reported [61,62], although the mechanisms responsible are not known. In recent years, progress has been made towards characterizing the genetic profile of WM tumor cells. Whole-genome sequencing has made it possible to identify a recurrent somatic mutation, the *MYD88* L265P, as a unifying event in most patients with WM (95–97%) and IgM-MGUS (90%) [63,64,65,66,67]. However, studies in mice have shown that although this alteration might be indispensable for the WM phenotype, it is insufficient by itself for the full development of lymphoma [68]. Whether it may be the tumor-initiating event that confers a competitive advantage on the clone and predisposes it to further genetic alterations remains to be clarified. A recent study has shown the presence of *MYD88* L265P in B-cell precursors from 6/10 patients and in residual normal B-cells from 6/10 patients [69]. It would be interesting to evaluate the different cell populations in Ig-MGUS and WM patients at the single-cell level to identify the event(s) that accompany or co-operate with *MYD88* mutation in the malignant transformation. 

The second most common alteration in WM affects the C-terminal domain of *CXCR4* and is present in 40% of patients, almost all of whom are *MYD88*-mutated [70,71]. Currently, more than 40 different mutations (either nonsense or frameshift) have been described, the most frequent being the nonsense *CXCR4* S338X mutation, which is present in ~25% of patients [72,73]. *CXCR4* mutations are mostly sub-clonal, with a variable clonal distribution, which suggests they are probably acquired after the *MYD88* mutation during WM oncogenesis [71,74,75]. One single-cell study has already addressed the differences between cells that acquire the *CXCR4* mutation and those that remain *CXCR4*-wild-type within a patient. Results did not show a distinct mutation profile for the two populations, but further studies are needed [76]. 

There is increasing evidence about the role of the B-cell receptor pathway in the pathobiology of WM, either alone or in cooperation with the MYD88 signaling axis [77,78,79,80,81,82,83]. Mutations in *CD79A* and *CD79B* occur in 8–12% of WM patients, and co-expression of *CD79B* and *MYD88* mutations has been associated with transformation to aggressive lymphoma [84,85,86,87]. Lastly, mutations in *TP53* are more rarely observed—they occur in 2–3% of WM—and are associated with poor survival (Figure 1) [77,85,88,89].

All this means that, in spite of the monoclonality inherent to all cancer cells, they harbor a substantial degree of variability that could be responsible for the clinical heterogeneity of WM. However, to date, no clear correlation has been found between the mutational profile and the clinical behavior of IgM gammopathies [84,85]. The global frequency of mutations and genomic alterations (copy-number abnormalities, loss of heterozygosity) does increase from IgM-MGUS to SWM, which confirms the association between aggressive clinical behavior and a higher frequency of alterations [84,85,86,90]. Therefore, it appears that these alterations could have a role in the multistep oncogenic process that drives the transition first from IgM-MGUS to AWM, and then to SWM. Considering the heterogeneity of the mutational profile of WM patients (Figure 1), it seems unlikely that the malignant clone has a specific pattern of aberrations. In fact, the evidence suggests that there may be at least two pathways that promote IgM-MGUS progression to symptomatic disease: *CXCR4* mutation and 6q21-25 deletion. Loss of chromosome 6q is found in 40–50% of patients with WM [91,92] and appears to be exclusive to *CXCR4* in treatment-naïve patients [93]. The increasing frequency of 6q deletions from IgM-MGUS through asymptomatic and symptomatic WM suggests that loss of genes within this region facilitates disease progression [84,91,93,94]. As for *CXCR4* mutations, the transcriptomic profile, relative to that of solely *MYD88-*mutated individuals, shows diminished B-cell differentiation, downregulation of tumor suppressors overexpressed by *MYD88* mutation, and alternative activation of the toll-like receptor 7 (TLR7) pathway [95].

As noted in other tumors, it is not only the type of genetic alteration that matters—the target cell where mutations arise and the order of the events are also of great relevance to the course of the oncogenic process. Therefore, it would be interesting to perform multi-omics analyses in single cells, distinguishing between the three stages of the disease (IgM-MGUS, AWM, and SWM), to resolve the zygosity and co-occurrence/exclusion of mutations, and to examine they extent to which they are correlated with the immunophenotype. With these insights, phylogenetic trees could be reconstructed to determine the order of acquisition of the alterations, the cell of origin, and light could be shed on the putative mechanisms leading to progression. The same strategy could be applied to the transformation from WM to DLBCL or leukemia. It is conceivable that some of the genetic abnormalities driving this process may already be present at diagnosis in certain subclones. Identifying these events would provide a comprehensive view of this transition and would help to develop new diagnostic strategies and targeted therapies. According to previous studies, the transformation process is consistent with a branching model of evolution in which only clones containing driver mutations give rise to more aggressive populations by acquiring new aberrations (Figure 2) [87]. Identical scenarios have been reported in MM [96], FL [97], chronic lymphocytic leukemia [98], AML [99], ALL [100], and even solid tumors [6], although these need to be confirmed in further analyses of single cells. Likewise, it would be interesting to establish whether there is a progenitor tumor cell that is common to both diseases (WM and DLBCL) and that is responsible for their pathogenesis.

The heterogeneous composition of the bone marrow infiltration in WM, including variable percentages of small lymphocytes, plasmacytoid lymphocytes, and plasma cells [101], is also amenable to study at the single-cell level. The pathophysiology of the plasma cell involvement is poorly understood, and it has been observed that residual plasma cells may be present even in the complete response when the neoplastic B-cell component is absent [102,103]. Moreover, according to a recent study, concordance between the mutational landscape of WM B-cells and plasma cells is not 100%, suggesting that not all WM B-cells differentiate into plasma cells [69]. With more in-depth knowledge of the different populations, it may be possible to determine whether they are part of the neoplastic clone and to identify the origin of the WM cell.

Another interesting outcome of single-cell technologies would be to determine therapy resistance mechanisms throughout the course of treatment. Ibrutinib, a Bruton tyrosine kinase (BTK) inhibitor, is highly active in WM patients [104]. Mutations in *BTK* at its binding site (Cys481) are common in patients who experience progression on ibrutinib, particularly those with *CXCR4* mutations. Multiple *BTK* mutations can occur within individual patients and coexist with *PLCγ2*-activating mutations, which are also associated with acquired ibrutinib resistance, both of which lead to downstream ERK1/2 activation [105]. In most cases, *BTK* mutations are sub-clonal but confer a protective effect against ibrutinib on neighboring BTK wild-type cells through a paracrine mechanism [106]. Single-cell studies may help address the pathways and mechanisms induced in wild-type cells by these alterations, as well as other treatment-emergent mutations, the reason for their co-occurrence, and the particular susceptibility of *CXCR4*-mutated cells. Such findings would enable us to identify patterns of clonal selection and evolution that mediate clinical resistance, and thereby to use an alternative strategy to overcome resistance in these patients.

It is important not to limit the studies solely to DNA. There may be no direct correspondence between the genetic findings and the final phenotype because important steps must be traversed from one to the other. Transcription and translation, together with all the key regulatory processes involved, play a fundamental role in the pathogenesis of the disease, and everything should be studied and interpreted as a whole. *MYD88* and *CXCR4* mutations divide WM patients into three major genetic subgroups (*MYD88^WT^*, *MYD88^L265P^*-*CXCR4^WHIM^*, and *MYD88^L265P^*-*CXCR4^WT^*) that show upregulation and downregulation of different pathways, which is translated into the three final clinical phenotypes with their distinct clinical features [95]. The most clinically relevant consequence of this subclassification of WM is the differential response to certain drugs, such as ibrutinib. Thus, patients with wild-type *MYD88* do not achieve an objective response to ibrutinib [107], while those with *CXCR4* mutations show lower rates of deep responses, and shorter progression-free survival [108,109]. Therefore, it is crucial that all the information be considered as a whole, and single-cell DNA sequencing studies must be complemented by studies of the phenotypic consequences that a specific genomic abnormality may imply.

## 4. Limitations of Single-Cell Analysis

As we have shown throughout this review, single-cell technologies have many applications and enormous potential, however, their limitations and challenges must also be recognized. First of all, a large quantity of fresh tumor cells of high viability and purity is essential, but we know that this requirement would be difficult to fulfill for certain diseases. For DNA and protein analysis, it is possible to freeze the samples until they are processed, but for single-cell RNA-sequencing, it is advisable to process the samples on the day of collection to minimize the risk of changes in expression. The workflow is not particularly complicated, but appropriate bioinformatic support and the use of computational methods are absolutely essential. The amount of information generated must be properly processed and analyzed by an integrated pipeline for multi-omics analysis. Allelic dropout, false genotype measurements, sequence-dependent bias, and PCR errors must be considered when analyzing single-cell DNA sequences [110]. For single-cell RNA sequencing, the problems that must be overcome are the poor detection of genes with low-level expression, the 3’ bias, and the noise in the transcriptome data [111]. Different strategies, such as fluorescence-activated cell sorting, Western blotting, metal-tagged antibodies, or oligonucleotide-labeled antibodies, can be used to obtain single-cell proteome profiles, but none of them can detect all the proteins that are expressed [112]. The combination of different single-cell genomic signals is computationally challenging, as these data are intrinsically heterogeneous for experimental, technical, and biological reasons. Multidimensional data obtained from a single cell are integrated and interpreted by computational methods, which provide accurate tools for analyzing multimodal data [113]. Software tools for sequence import and data analysis and visualization are very useful, and some companies already provide them with their products. The great complexity of the data means that results must be interpreted with caution and no clinical decision should be made based on them.

Another limitation of single-cell studies is their expense. As with any emerging technique, up-front costs for consumables, labor, and sequencing are high, creating a barrier to widespread implementation, and excluding these technologies from use in routine testing. Developing high-throughput technologies with a high sample-processing capacity along with easily fabricated and conducted processes could lower the price. The automation of devices would also reduce assay time and minimize human intervention, in addition to limiting user bias and improving reproducibility across laboratories [114].

## 5. Conclusions

In summary, single-cell analysis can overcome the limitations of bulk sequencing and provide unique insights into tumor heterogeneity, co-dependency of malignant events, and phylogenesis at the cellular level. Furthermore, single-cell multi-omics makes it possible to carry out high-dimensional studies to address complex biological questions, by simultaneously combining the genomic, transcriptomic, and proteomic profiles from the same cell. The integrated information can be used to select the best strategy for the management and treatment of patients. 

## Figures and Tables

**Figure 1 cancers-13-01541-f001:**
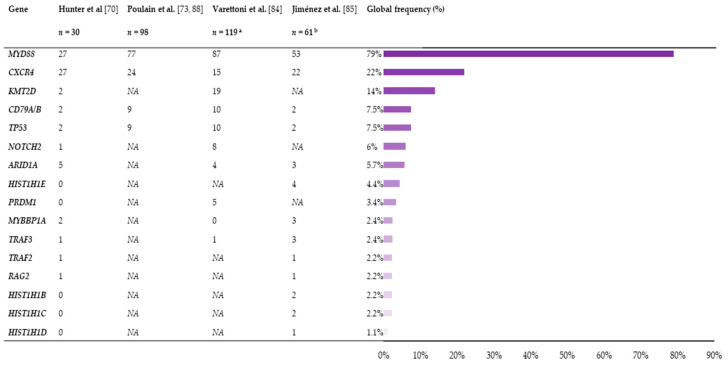
Frequency and distribution of the mutations in the main genes of Waldenström’s macroglobulinemia (WM) according to the different studies. The total number of mutated patients in each study, as well as the global mutation frequency (%) considering the four studies, are displayed. ^a^ Corresponding to 57 Immunoglobulin M monoclonal gammopathy of undetermined significance (IgM-MGUS) and 62 WM; ^b^ Corresponding to 14 IgM-MGUS, 23 asymptomatic Waldenström’s macroglobulinemia (AWM) and 24 symptomatic Waldenström’s macroglobulinemia (SWM). NA: not applicable.

**Figure 2 cancers-13-01541-f002:**
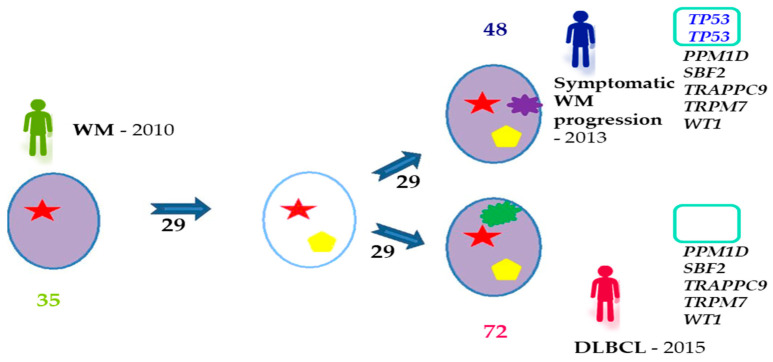
Branching model of tumor evolution observed in WM transformation to DLBCL, adapted from Jiménez, C., Alonso-Álvarez, S., Alcoceba, M. et al. From Waldenström’s macroglobulinemia to aggressive diffuse large B-cell lymphoma: a whole-exome analysis of abnormalities leading to transformation. Blood Cancer J. 7, e591 (2017). https://doi.org/10.1038/bcj.2017.72 [87]. The figure shows an example from a patient who was diagnosed with WM in 2010 and who transformed to DLBCL in 2015, with a symptomatic progression in 2013 before the transformation: 35 mutations were identified at diagnosis, 48 at relapse and 72 at transformation, and 29 of the alterations were conserved throughout the entire process. Mutations in *PPM1D*, *SBF2*, *TRAPPC9*, *TRPM7,* and *WT1* genes were present at progression and transformation. By contrast, two *TP53* mutations at relapse that were not observed at transformation. This implies that the transformed final clone did not evolve from the same subclone as was responsible for progression, but from a previous one that would not yet have acquired the *TP53* mutations. DLBCL, diffuse large B-cell lymphoma; WM, Waldenström’s macroglobulinemia.

**Table 1 cancers-13-01541-t001:** Summary of the main applications of single-cell technologies in different areas of hematology. CAR T, chimeric antigen receptor T-cell; Cas9, CRISPR-associated endonuclease Cas9; CRISPR, clustered regularly interspaced short palindromic repeats.

Area	Application	Reference
Immune system	Study stem cell-like CD8+ memory T cells to develop immunotherapies and vaccines	[21]
Tumor cells	Combine genotyping and immunophenotyping to fully characterize the disease	[23]
Clonal hematopoiesis	Distinguish mutations associated with clonal hematopoiesis vs. true leukemia to accurately measure residual disease	[29]
Oncogenesis	Establish the sequence of genetic events that occur in the disease development; characterize mutational combinations that promote clonal expansion to select targeted therapies	[32,37]
Clonal evolution	Study the predisposition to develop leukemia in Shwachman-Diamond syndrome to identify patients with high-risk clones	[40]
Therapy resistance	Characterize clonal complexity to predict clinical relapse; evaluate concurrence of different resistance mechanisms to search for novel treatment strategies	[35,43,46,47]
Microenvironment	Define the supportive role of the immune microenvironment to develop new therapeutic approaches	[51]
B-cell biology	Model the germinal center to understand lymphoma pathology	[53]
Cancer therapy	Identify transcriptomic features in anti-CD19 CAR T-infusion cell products to determine efficacy and toxicity	[55]
CRISPR-Cas9 gene editing	Analyze Cas9-introduced gene edits to quantify the abundance of CRISPR-introduced disease drivers and decipher the effects of multiplex gene editing	[56]
Bone marrow transplant	Assess donor/host chimerism to monitor bone marrow engraftment and predict relapse after transplant	[57]

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
