# Peer review of "Time to Move to the Single-Cell Level: Applications of Single-Cell Multi-Omics to Hematological Malignancies and Waldenström’s Macroglobulinemia—A Particularly Heterogeneous Lymphoma"

_cancers, 2021, doi:10.3390/cancers13071541_

Round 1

Reviewer 1 Report

Garcia-Sanz and Jimenez present a review on single cell studies in hematology , and then focus on potential applications in one disease, WM. This is a timely review, and the authors have the expertise to put the data into perspectives. They succeed in organizing the magnitude of studies on very divergent topics into praragraphs adressing different biological pathways.

The cited refernces will help the reader for further reading on techniques and single topics.

Minor comment:

line 134: it should probablly  read “are present “instead of “represent”

Author Response

We thank the reviewer for the comments and have modified line 134 of the manuscript accordingly:

'Presence of clonal hematopoiesis has also been assessed by single cell DNA sequencing in patients with myelodysplastic syndrome-associated phenotypic abnormalities, which are present in ∼10% of newly diagnosed multiple myeloma (MM) [31].'

Reviewer 2 Report

This review is well written and unravels several applications of the single cell analysis in hematology and, in particular, in Waldenström’s Macroglobulinemia. While the practical aspects of these methods are clear and well developed, the authors quit to discuss the technical aspects of these technologies. To spend a few words more may help the not-expert readers to understand the huge potential of such approach.

The issue treated is a very important topic of the actual research about hematological malignancies, but as authors stated, some limitations may hinder a wide use in current molecular diagnostics to drive therapeutic strategies.

Author Response

Answer: We agree with the reviewer and we have included the following paragraph in the manuscript:

'Single-cell methods involve single-cell isolation, barcoding and sequencing to analyze multiple types of molecules from individual cells, as well as the integrative analysis of the data to characterize cell types and their functions based on molecular signatures. To analyze multiple types of molecules from the same cell it is essential to isolate the single-cells and then to barcode the molecules. The process will be explained in greater details for two of the most widely used single-cell systems. The Tapestri Platform (Mission Bio) allows DNA and protein analysis of each individual cell and is based on a two-step microfluidic workflow. The first step involves encapsulating the cells into sub-nanoliter droplets and isolating DNA and oligo-conjugate antibodies from each single cell. The second step involves cell barcoding and targeted amplification by PCR within the droplets, which are then disrupted, before extracting barcoded DNA for library amplification. Final libraries are purified and sequenced. The Chromium Platform (10x Genomics) enables the analysis of gene expression, chromatin accessibility, cell-surface proteins, immune clonotype, antigen specificity and CRISPR edits. It is based on the generation of thousands of single-cell partitions, each containing an identifying barcode for downstream analysis using advanced microfluidics. Within the instrument, barcoded gel beads, coated with a unique oligonucleotide barcode sequence and functionalized sequences to capture molecules of interest, are mixed with cells or nuclei, enzymes, and partitioning oil to form thousands of single-cell emulsion droplets. Each droplet is the location of an individual reaction in which the beads are dissolved and molecules of interest from each cell are captured and barcoded. After barcoding, all fragments from the same cell or nucleus share a common barcode. Barcoded fragments from thousands of cells are pooled for downstream reactions to create sequencing libraries. After sequencing, bioinformatics tools use the identifying barcodes to map sequencing reads back to their single cell or nucleus of origin.'

Reviewer 3 Report

As far as I can tell it is a balanced and not-so-hard-to-read review.

It would be informative to write out the antibody targets for the biologicals that get mentioned, e.g. that blinatumomab is anti-CD4/CD19, and also more generic since the macromolecules are producer-specific. I can see it's already done for one of them.

"Advances in single cell sequencing have streamlined this process; in particular, single cell VDJ sequencing data is used to clone antigen‐specific TCRs. The same strategy can be used to expand primary human antigen‐specific T cell clones [54]" As the source says, this still is still hard work, it's not all that streamlined

"For DNA and protein analysis, it is possible to freeze the samples until they are to be processed, but for single cell RNA sequencing it would be advisable to process the samples in the same day of collection, since changes in expression may occur." Yes, but once extracted RNA can be frozen

"problems to overcome are the poor detection of low‐expressed genes, the 3’ bias, the transcript dropout" Here "poor detection of low‐expressed genes" and "transcript dropout" are basically the same thing

Author Response

As far as I can tell it is a balanced and not-so-hard-to-read review.

It would be informative to write out the antibody targets for the biologicals that get mentioned, e.g. that blinatumomab is anti-CD4/CD19, and also more generic since the macromolecules are producer-specific. I can see it's already done for one of them.

Answer: This is an interesting remark and we have now included that information in the paper:

  • ‘In patients with AML treated with combinations based on venetoclax (inhibitor of BCL-2), NPM1 or IDH2 mutations are associated with high response rates and durable remissions, while the activation of FLT3, RAS or TP53 signaling pathways seems to be linked to resistance development [45].’
  • ‘The complex biology of resistance was also underscored in AML treated with ivosidenib (an inhibitor of mutated IDH1), as the concurrence of different mechanisms, particularly receptor tyrosine kinase pathway mutations and IDH-related mutations, contributed to primary and secondary resistance [47].’
  • ‘By RNA sequencing, a CD19 splice variant was identified as a biomarker for failure to blinatumomab (anti-CD3/CD19) therapy in B-cell progenitor ALL [48], and a USP7 gene signature was associated with resistance to therapy in AML patients [49].’
  • ‘Lastly, a deeper analysis of the CLL response to ibrutinib (a BTK inhibitor) allowed to dissect the precise cellular and molecular changes induced by this drug and to identify candidate molecular markers of therapy response.’

"Advances in single cell sequencing have streamlined this process; in particular, single cell VDJ sequencing data is used to clone antigen‐specific TCRs. The same strategy can be used to expand primary human antigen‐specific T cell clones [54]" As the source says, this still is still hard work, it's not all that streamlined

Answer: That is true and the following modification has been made in accordance:

‘Advances in single cell sequencing have begun to streamline this process; in particular, single cell VDJ sequencing data is used to clone antigen‐specific TCRs.’

"For DNA and protein analysis, it is possible to freeze the samples until they are to be processed, but for single cell RNA sequencing it would be advisable to process the samples in the same day of collection, since changes in expression may occur." Yes, but once extracted RNA can be frozen

Answer: This is a right observation but for single cell sequencing, libraries are straightly prepared from cell or nuclei suspensions, there is no DNA or RNA extraction step.

"problems to overcome are the poor detection of low‐expressed genes, the 3’ bias, the transcript dropout" Here "poor detection of low‐expressed genes" and "transcript dropout" are basically the same thing

Answer: We agree with the reviewer and we have modified the manuscript accordingly:

‘For single cell RNA sequencing, the problems to overcome are the poor detection of low‐expressed genes, the 3’ bias and the noise in the transcriptome data [113].’

Reviewer 4 Report

This review is well presented and in depth.  Some grammatical changes are required  for readability. 

Author Response

We thank the reviewer for the suggestion and we have sent the manuscript to an English proofreader to edit the English language and style.